

# Plant growth regulators mitigate oxidative damage to rice seedling roots by NaCl stress

Yaxin Wang[1], Li-ming Zhao[1], Naijie Feng[1,2,3], Dianfeng Zheng[1,2,3], Xue Feng Shen[1,2,3], Hang Zhou[1,2], Wenxin Jiang[1], Youwei Du[1], Huimin Zhao[1], Xutong Lu[1] and Peng Deng[1]

[1] College of Coastal Agricultural Sciences, Guangdong Ocean University, Zhanjiang, Guangdong, China
[2] National Saline-tolerant Rice Technology Innovation Center, South China, Zhanjiang, Guangdong, China
[3] Shenzhen Institute of Guangdong Ocean University, Shenzhen, Guangdong, China

Corresponding authors
Li-ming Zhao, nkzlm@126.com
Naijie Feng, fengnj@gdou.edu.cn

## ABSTRACT

The aim of this experiment was to investigate the effects of exogenous sprays of 5-aminolevulinic acid (5-ALA) and 2-Diethylaminoethyl hexanoate (DTA-6) on the growth and salt tolerance of rice (*Oryza sativa* L.) seedlings. This study was conducted in a solar greenhouse at Guangdong Ocean University, where 'Huanghuazhan' was selected as the test material, and 40 mg/L 5-ALA and 30 mg/L DTA-6 were applied as foliar sprays at the three-leaf-one-heart stage of rice, followed by treatment with 0.3% NaCl (W/W) 24 h later. A total of six treatments were set up as follows: (1) CK: control, (2) A: 40 mg· L$^{-1}$ 5-ALA, (3) D: 30 mg· L$^{-1}$ DTA-6, (4) S: 0.3% NaCl, (5) AS: 40 mg· L$^{-1}$ 5-ALA + 0.3% NaCl, and (6) DS: 30 mg· L$^{-1}$ DTA-6+0.3% NaCl. Samples were taken at 1, 4, 7, 10, and 13 d after NaCl treatment to determine the morphology and physiological and biochemical indices of rice roots. The results showed that NaCl stress significantly inhibited rice growth; disrupted the antioxidant system; increased the rates of malondialdehyde, hydrogen peroxide, and superoxide anion production; and affected the content of related hormones. Malondialdehyde content, hydrogen peroxide content, and superoxide anion production rate significantly increased from 12.57% to 21.82%, 18.12% to 63.10%, and 7.17% to 56.20%, respectively, in the S treatment group compared to the CK group. Under salt stress, foliar sprays of both 5-ALA and DTA-6 increased antioxidant enzyme activities and osmoregulatory substance content; expanded non-enzymatic antioxidant AsA and GSH content; reduced reactive oxygen species (ROS) accumulation; lowered malondialdehyde content; increased endogenous hormones GA3, JA, IAA, SA, and ZR content; and lowered ABA content in the rice root system. The MDA, $H_2O_2$, and $O_{2-}$ contents were reduced from 35.64% to 56.92%, 22.30% to 53.47%, and 7.06% to 20.01%, respectively, in the AS treatment group compared with the S treatment group. In the DS treatment group, the MDA, $H_2O_2$, and $O_{2-}$ contents were reduced from 24.60% to 51.09%, 12.14% to 59.05%, and 12.70% to 45.20%. In summary, NaCl stress exerted an inhibitory effect on the rice root system, both foliar sprays of 5-ALA and DTA-6 alleviated damage from NaCl stress on the rice root system, and the effect of 5-ALA was better than that of DTA-6.

## INTRODUCTION

In recent years, land resources have been degraded, soil organic matter and fertility have declined, and salinity has increased severely due to increased human activities and irrational farming practices (*Shahid, Zaman & Heng, 2018*; *Page et al., 2021*). More than one-third of the world's irrigated land is affected by salinity, and salt stress has become one of the most significant abiotic stresses affecting agricultural production (*Dubois, 2011*; *Zhao et al., 2020a*). Salt stress exposes plant growth and developmental processes to osmotic stress hazards, which reduces water availability and the high accumulation of reactive oxygen species (ROS), thereby inhibiting plant growth (*Soni et al., 2021*). Therefore, improving plant salt tolerance has become essential for further increasing agricultural production and income.

Rice (*Oryza sativa* L.) is a staple food crop for half of the world's population and is widely distributed in places such as Asia, Latin America, and Africa. Rice cultivation is essential to ensure global food security (*Gommers & Monte, 2018*; *Hussain et al., 2017*). There is about 1 billion hm$^2$ of saline-alkali land in the world, and countries are vigorously developing and using coastal mudflats to develop rice production. Rice seedling establishment is a complex process of environmental stimulation and phytohormone regulation as well as a critical developmental step in the continuation of the rice life cycle (*Zhao et al., 2020b*; *He et al., 2020*; *Xu et al., 2020*). Rice seedling root systems are more salt-sensitive than other cereals, and salt stress can lead to a decrease in soil solution in rice roots, resulting in cell water loss and even plant death (*Krishnamurthy et al., 2016*).

Plant growth regulators (PGRs) have been found to enhance abiotic stress tolerance in plants. Previous studies have found that PGRs play an important role in the interaction between plants and the environment, and are known as the coordinators of stress response and plant growth (*Bueno & Cordovilla, 2021*; *Slama et al., 2020*). When comparing factors such as stress resistance breeding, resistance genetics, and fundamental improvement of soil salinity of rice varieties, the exogenous spraying of PGRs not only improves the salt tolerance of rice, but also has the characteristics of low cost, simple operation, and economic efficiency, which has received extensive attention in recent years.

5-Aminolevulinic acid (5-ALA) is one of the most critical plant growth regulators and essential precursors for heme and chlorophyll synthesis (*Hotta et al., 1997*). 5-ALA has been shown to regulate plant growth and development, increase chlorophyll content, promote photosynthesis, and reduce membrane lipid peroxidation damage (*Akram, Ashraf & Al-Qurainy, 2012*). A previous study used 5-ALA to enhance the photosynthetic rate and thus growth in barley, rice, garlic, and potato (*Hotta et al., 1997*). In addition, low concentrations of 5-ALA have been shown to regulate the physiological properties of plant growth under abiotic stresses such as salinity (*Youssef & Awad, 2008*), low temperature (*Akram & Ashraf, 2013b*; *Balestrasse et al., 2010*), heavy metals (*Ali et al., 2014*), and drought (*Al-Thabet, 2006*). However, 5-ALA's mechanism in rice root growth under salt stress has not been fully elucidated.

Diethyl aminoethyl hexanoate (DTA-6) is a novel cytokinin plant growth regulator (*He, Wu & He, 2013*) that is widely used in cabbage, cotton, corn, wheat, soybean, and amaranth

 

other crops (*Jiang et al., 2012*; *Li & Lin, 2020*). It has been shown that DTA-6 can increase the activity of antioxidant enzymes, increase chlorophyll content, promote photosynthesis, promote cell division and elongation, and regulate nutrient balance in the body (*Youssef & Awad, 2008*). Exogenous DTA-6 has also been applied to improve the quality and quantity of microalgal lipids for biodiesel production (*Salama et al., 2014*). In addition, exogenous DTA-6 enhanced fatty acid and sugar metabolism in soybean, as well as promoted seed germination and seedling growth in aged soybean (*Zhou et al., 2019*). DTA-6 also plays an essential role in regulating abiotic stresses such as temperature, drought (*Yu et al., 2008*), heavy metals (*He, Wu & He, 2014*), and salinity. To date, the physiological mechanisms of DTA-6 on rice root growth under salt stress have not been investigated.

In China, there are large areas of rice cultivation that are threatened by salt stress, and their productivity is severely affected by the accumulation of soluble salts in the soil. It is essential to enhance the physiological mechanisms of the rice root system through PGRs to improve the salt tolerance of rice seedlings. There have been few reports on the physiological regulation effect of 5-ALA and DTA-6 foliar sprays on rice, and the effect of these two plant growth regulators on salt stress in rice roots has not been reported. Therefore, this experiment was conducted to investigate the effects of exogenously applied 5-ALA and DTA-6 on rice root morphology, root vigour, degree of membrane damage, antioxidant enzyme activities, osmoregulatory substances, and endogenous hormones, with an aim of providing a theoretical basis for the application of the two PGRs in salt-resistant rice cultivation.

## MATERIAL AND METHODS

### Materials and reagents

The experiment was conducted in the summer of 2021–2022 in a solar greenhouse in the Binhai Agricultural College of Guangdong Ocean University. The average indoor temperature was $33 \pm 2$ °C during the day and $30 \pm 2$ °C at night, and the humidity was controlled at $80\% \pm 5\%$. The test variety was Huanghuazhan, selected by the Rice Research Institute of Guangdong Academy of Agricultural Sciences (Shenzhen Longping Jingu Seed Co., Ltd.). The test regulators were 5-ALA (Aladdin, Beijing, China) and DTA-6 (Zhengzhou Xinlian Biochemical Technology Co., Ltd, Zhengzhou, China). The soil used for the test was a 3:1 mixture of brick-red soil and river sand. The physicochemical properties of the soil were: pH 7.02, organic matter 3.52 g/kg, alkali-hydrolyzable nitrogen 1.26 mg/kg, quick-acting phosphorus 2.62 mg/kg, and quick-acting potassium 83.39 mg/kg.

### Experimental designs

We selected rice seeds that were full and intact, disinfected them with 3% hydrogen peroxide for 15 min, and rinsed them with distilled water five times. The seeds were first soaked in distilled water for 24 h and germinated for 24 h. Both soaking and germination were carried out in the dark at 30 °C in a thermostat. One day before sowing, the bottom fertilizer was poured into non-porous flower pots with an upper diameter of 20 cm, lower diameter of 15 cm, and height of 18 cm. Subsequently, the seeds were sown in non-porous pots. Each pot held 3 kg of soil, and 69 seeds were evenly sown in each pot, with a plant

spacing of 0.5 cm. We watered the bottom fertilizer 1 day before sowing. After the seedlings were subjected to regular water management until the three-leaf-one-heart stage, 5-ALA and DTA-6 foliar sprays were applied in doses of 40 mg· L$^{-1}$ and 30 mg· L$^{-1}$, respectively, around 19:00 after dark (*Li et al., 2021*), and clear water was used as a control. Salt treatment was carried out after 24 h of conditioner treatment, *i.e.,* 0.3% NaCl (W/W) was applied according to the soil volume, and 9 g of salt dissolved in 1 L of water was slowly poured into the pots to simulate the field trial, retaining a two cm water layer. Salt concentration was maintained daily by real-time salinity monitoring with a soil salinometer. The treatments of the experiment were (1) CK: control, (2) A: 40 mg· L$^{-1}$ 5-aminolevulinic acid (5-ALA), (3) D: 30 mg· L$^{-1}$ 2-Diethylaminoethyl hexanoate (DTA-6), (4) S: 0.3% NaCl, (5) AS: 40 mg· L$^{-1}$ 5-ALA + 0.3% NaCl, and (6) DS: 30 mg· L$^{-1}$ DTA-6 + 0.3% NaCl. With three replications of 10 pots per treatment and a total of 30 pots per treatment, the trial was a completely randomized placement design. Samples were taken at 1, 4, 7, 10, and 13 d after salt treatment to determine the indicators. Sampling was divided into two parts: the rice plants of each treatment were arranged in order, and the soil around the roots was rinsed with clean water until it was completely clean. Some rice plants were used to determine root morphological indexes. Next, we absorbed the surface moisture of the root system with absorbent paper, separated the aboveground part and the underground part, and quickly put the underground part into liquid nitrogen for the determination of physiological indicators.

### Determination of morphological indicators in the subsoil

First, we collected the rice seedlings from each treatment separately and rinsed them with water. Second, 15 representative seedlings were selected, and the aboveground and underground parts were separated. Next, we placed the rice roots horizontally and straightened them naturally. Finally, the root length was measured with a graduated ruler and its value was recorded manually. After the underground sample was packed, it was put into the oven at 105 °C for 30 min, dried in the 80 °C electric blast drying oven for 48 h to constant weight, and the mass of the underground dry matter was weighed with an electronic scale, and its value was manually recorded.

### Determination of membrane damage index

Determination of malondialdehyde (MDA) was conducted using the thiobarbituric acid method (*Chen et al., 2022a*). The hydrogen peroxide (H$_2$O$_2$) content was determined according to *Jessup, Dean & Gebicki (1994)*. The rate of superoxide anion production (O$_2$·$^-$) was determined according to the method described by *Lin et al. (2020)*.

### Measurement of physiological and biochemical indicators

The crude enzyme was extracted as described by *Lee & Lee (2000)*. We weighed 0.5 g of fresh rice leaves, homogenized with 10 mL (pH 7.8) of sodium phosphate buffer (PBS), and centrifuged for 15 min at 10,000 rpm (revolutions per minute) at 4 ° C to obtain the crude enzyme extract, with three replicates per treatment. Superoxide dismutase (SOD) activity was calculated using the nitrotetrazole (NBT) method described by *MacAdam, Nelson & Sharp (1992)*. Peroxidase (POD) activity was determined according to the guaiacol method

described by *MacAdam, Nelson & Sharp (1992)*. Catalase (CAT) activity was determined according to the hydrogen peroxide method described by *Li et al. (2019)*. Ascorbate peroxidase (APX) activity was determined according to the method described by *Nakano & Asada (1981)*.

The AsA content was measured according to the method of *Chen et al. (2022b)*. The absorbance value was read at 534 nm. Reduced glutathione (GSH) content was determined according to the method described by *Ellman (1959)*, and soluble protein (SP) content was determined according to the method of *Bradford (1976)*.

Soluble sugar (SS) content was determined using the colorimetric method described by *Zhang et al. (2017)* with slight modifications. Briefly, 0.2 g of sample and eight mL of distilled water were placed in a test tube, which was placed in a boiling water bath for 30 min and allowed to cool. Then, 0.5 mL of ethyl ketone acetate and five mL of sulfuric acid were mixed, placed in a boiling water bath for 10 min, and allowed to cool. The light absorbance value of each sample was measured at 630 nm using a UV/Vis spectrophotometer. Sugar content was determined using a standard linear equation and then calculated in mg· $g^{-1}$. The sugar content of the samples was determined using a standard linear equation.

$$SS \; content(mg/g) = (C * V/a * n)/(W * 106) \tag{1}$$

C is the standard equation to obtain the amount of sugar (μg), a is the volume of aspirated sample (ml), V is the amount of extraction solution (ml), n is the dilution factor, and W is the weight of the tissue (g).

Shanghai Enzyme-linked Bioscience and Technology Co. determined the endogenous hormones. The kit was determined by double antibody one-step sandwich enzyme-linked immunosorbent assay (ELISA), and the absorbance (OD) was measured at 450 nm with an enzyme marker to calculate the concentration of the samples.

## Statistical analyses

Data were processed using Microsoft Excel 2010, one-way analysis of variance (ANOVA) was performed using SPSSX9 software, multiple comparisons were performed using Duncan's Multiple Test (Duncan's Test), correlation analysis was graphed with Origin2021 analysis, and other charts in the article were plotted with Origin2021.

## RESULTS

### Effects of PGRs on root morphology and root viability in rice

As shown in Table 1 and Fig. 1, foliar spraying of both PGRS increased root length and dry weight of root of rice under no NaCl stress. NaCl stress reduced root length and underground dry weight of rice seedlings. From 4 to 13 d after NaCl stress, root length and dry weight were lower than the control treatment, which showed a decrease of 2.54%–9.52% and 10.63%–23.37%, respectively, in the S treatment compared with the CK treatment. Under NaCl stress, the root length and dry weight of root AS treatment were higher than those of S. The increases in root length and dry weight of root of AS treatment over S treatment ranged from 2.20%–14.51% and 7.18%–63.11%, and the increase in root length and dry weight of root of DS treatment over S treatment ranged from 3.08% to 12.44%

**Table 1  Effects of PGRs on root length and shoot dry weight of rice at the tillering stage under salt stress.**

| Pattern indicator xes | Treatment | Salinity treatment time (d) | | | | |
|---|---|---|---|---|---|---|
| | | 1 | 4 | 7 | 10 | 13 |
| Root length/(cm) | CK | 18.80 ± 0.46a | 19.60 ± 0.61b | 19.73 ± 0.73cd | 20.50 ± 0.36d | 23.33 ± 0.33bc |
| | A | 19.70 ± 0.46a | 22.43 ± 0.12a | 24.43 ± 0.09a | 27.57 ± 0.15a | 28.07 ± 0.30a |
| | D | 18.97 ± 0.73a | 21.83 ± 0.17a | 22.63 ± 0.09b | 23.50 ± 0.289b | 24.0 ± 0.15b |
| | S | 17.53 ± 0.35a | 17.73 ± 0.63c | 18.83 ± 0.17d | 19.30 ± 0.25e | 22.73 ± 0.12c |
| | AS | 18.60 ± 1.45a | 18.93 ± 0.62bc | 20.97 ± 0.23c | 22.10 ± 0.31c | 23.23 ± 0.12bc |
| | DS | 18.07 ± 0.67a | 18.80 ± 0.21bc | 19.60 ± 0.62d | 21.70 ± 0.31c | 22.70 ± 0.31c |
| Root dry weight/($\times 10^{-2}$ g) | CK | 1.03 ± 0.03ab | 1.13 ± 0.09c | 2.07 ± 0.09d | 2.57 ± 0.09d | 3.13 ± 0.09c |
| | A | 1.10 ± 0.06a | 1.93 ± 0.09a | 3.13 ± 0.15a | 3.37 ± 0.07a | 4.37 ± 0.12a |
| | D | 1.13 ± 0.03a | 1.60 ± 0.10b | 2.57 ± 0.09bc | 2.86 ± 0.07c | 3.90 ± 0.15b |
| | S | 0.93 ± 0.07b | 0.93 ± 0.03c | 1.77 ± 0.09e | 1.97 ± 0.09e | 2.80 ± 0.06d |
| | AS | 1.00 ± 0.06ab | 1.50 ± 0.06b | 2.83 ± 0.09b | 3.20 ± 0.06ab | 4.57 ± 0.09a |
| | DS | 1.03 ± 0.03ab | 1.50 ± 0.06b | 2.50 ± 0.06c | 2.97 ± 0.09bc | 4.37 ± 0.07a |

**Notes.**
Data represents the means ± standard error. Different lowercase letters in the same column indicate a significant difference in Duncan at $P < 0.05$. CK (distilled water + 0% NaCl), S (distilled water + 0.3% NaCl), A (40 mg $L^{-1}$ 5-ALA + 0% NaCl) , AS (40 mg $L^{-1}$ 5-ALA + 0.3% NaCl), D (30 mg $L^{-1}$ DTA-6 + 0% NaCl) , DS (30mg $L^{-1}$ DTA-6+ 0.3% NaCl).

and from 10.72%–61.29% at 1–10 d after NaCl stress. In summary, both PGRs promoted rice seedling growth, with 5-ALA being superior to DTA-6.

NaCl stress decreased rice root vigor, and foliar spraying of both PGRs increased rice root vigor (Fig. 2). At 7 d after NaCl stress, the root vigor of both NaCl treatments was lower than that of the no-NaCl treatment, as shown by a 57.59% decrease in root vigor of the S treatment compared with that of the CK treatment. The root vigor of the A and D treatments was higher than that of the CK treatment, with an increase of 152.43% and 48.62%, respectively. Under NaCl stress, the root viability of both AS and DS treatments was significantly higher than that of S treatment, increasing by 152.47% and 113.73%, respectively. In summary, both PGRs promoted root growth in rice seedlings, with 5-ALA being superior to DTA-6.

## Effects of PGRs on membrane damage in rice roots

As shown in Fig. 3, foliar spraying of both PGRS reduced rice MDA, $H_2O_2$, and $O_2 \cdot^-$ content under no NaCl stress. NaCl stress increased rice seedlings' MDA, $H_2O_2$, and $O_2 \cdot^- \cdot$ content. MDA and $H_2O_2$ were higher than the control treatments from 1 to 13 d after NaCl stress, as evidenced by an increase of 12.57%–21.82%, 18.12%–63.10%, and 7.17%–56.20% in the S-treatment compared with the CK-treatment, respectively. The MDA and $H_2O_2$ contents of both AS and DS treatments were significantly lower than those of S treatment under NaCl stress. MDA, $H_2O_2$, and $O_2 \cdot^-$ content reduced by 35.64%–56.92%, 22.30%–53.47%, and 7.06%–20.01% in AS treatment compared to S treatment, respectively. The DS treatment MDA, $H_2O_2$, and $O_2 \cdot^-$ content decreased by 24.60% to 51.09%, 12.14%–59.05%, and 12.70%–45.20% compared to S treatment, respectively. Both PGRs showed significant effects in promoting rice seedling growth,

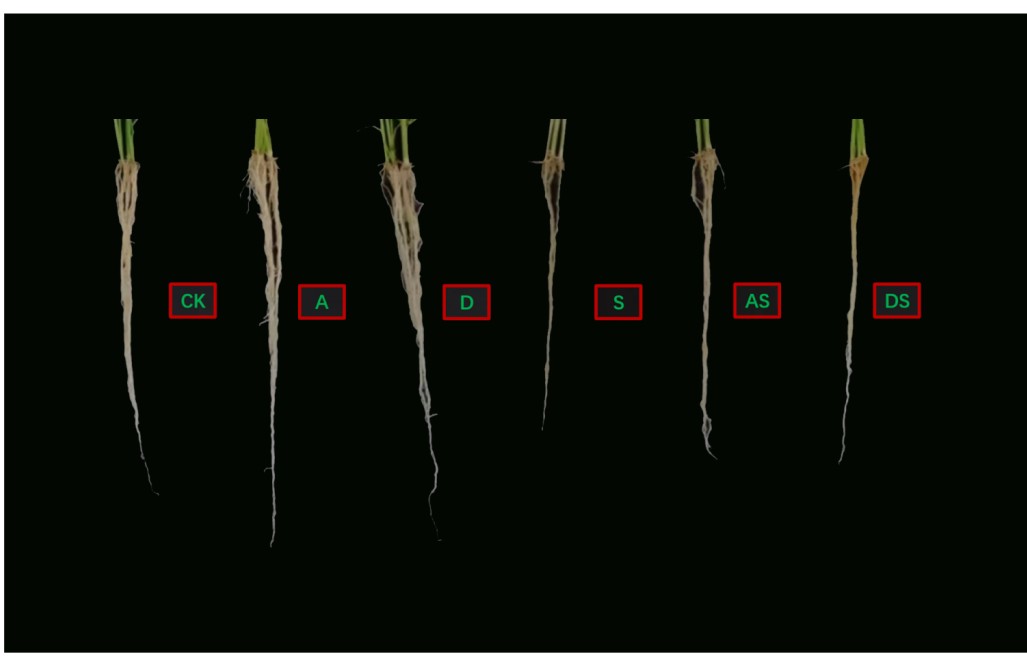

**Figure 1** **Effects of PGRs on the growth of rice seedings under NaCl stress (7 d).** Pictures show three seedlings per treatment. CK (distilled water + 0% NaCl), S (distilled water + 0.3% NaCl), A (40 mg L 5-ALA $^{-1}$ + 0% NaCl), AS (40 mg L $^{-1}$ 5-ALA + 0.3% NaCl), D (30 mg L $^{-1}$ DTA-6 + 0% NaCl), DS (30 mg L $^{-1}$ DTA-6+ 0.3% NaCl).

especially in alleviating the damage caused by NaCl stress in rice, with DTA-6 being more effective than 5-ALA.

## Effects of PGRs on antioxidant substances in rice roots

Foliar spraying of both PGRs increased SOD, POD, APX, and CAT activities in rice roots, with 5-ALA being superior to DTA-6 in enhancing antioxidant enzyme activities of roots (Fig. 4). In the absence of NaCl stress, SOD, POD, APX, and CAT activities were higher in A and D treatments than in the control group. NaCl stress significantly reduced SOD activity by 9.87% to 25.11%, and POD, APX, and CAT activities were significantly increased by 33.36%–45.96%, 30.50%–46.35% and 20.63%–41.41%, respectively, under NaCl stress (S) as compared to control (CK). Under NaCl stress, SOD, POD, APX, and CAT activities of AS treatment were significantly higher than those of S treatment, with increases ranging from 12.26%–59.15%, 20.97%–33.91%, 23.96%–56.96%, and 25.89%–89.53%, respectively. SOD, POD, and CAT activities significantly increased from 10.36%–56.21%, 13.60%–26.31%, and 16.58%–73.24% in DS treatment compared to S treatment, respectively.

In the absence of NaCl stress, foliar spraying of both PGRs increased AsA and GSH contents in rice roots (Figs. 4E and 4F). The A and D treatments increased AsA content compared to the CK treatment from 16.05%–40.32% and 5.27%–24.66%, respectively. GSH content increased from 55.79%–84.77% and 10.12%–54.06% in A and D treatments, respectively, compared to CK treatment. NaCl stress caused AsA and GSH to show a

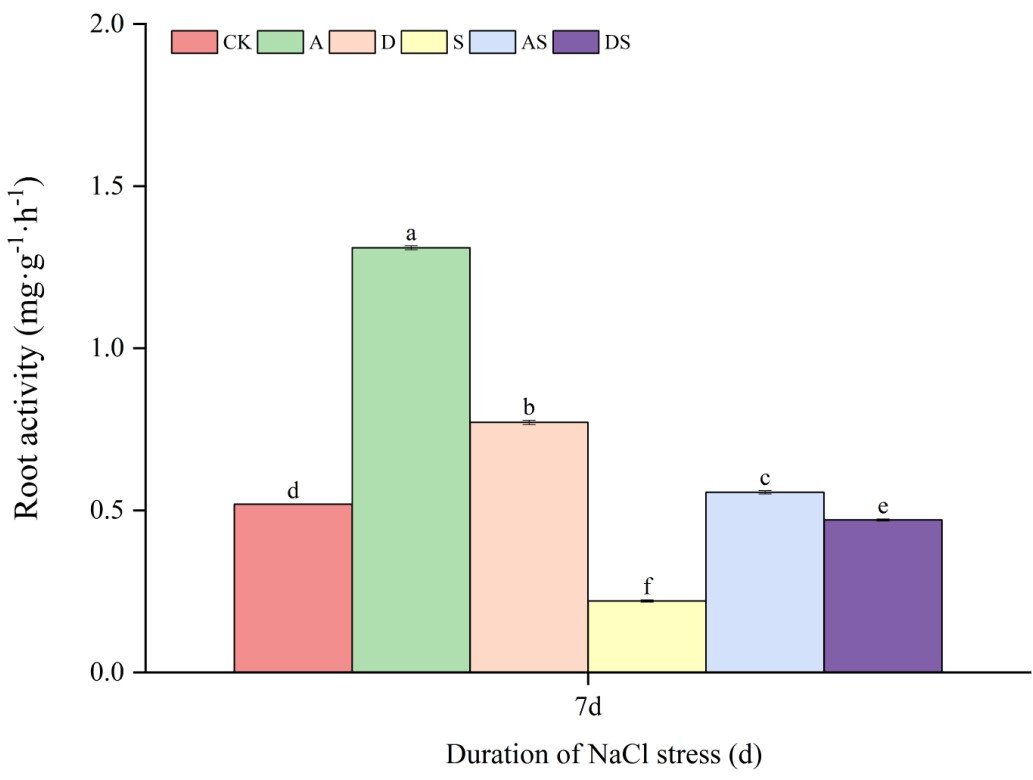

**Figure 2  Effects of PGRs on root vigor in rice.** Values are means ± SD ($n = 3$), and bars indicate SD. Columns with different letters indicate a significant difference at $P < 0.05$ (Duncan's test). CK (distilled water + 0% NaCl), S (distilled water + 0.3% NaCl), A (40 mg L$^{-1}$ 5-ALA + 0% NaCl), AS (40 mg L$^{-1}$ 5-ALA + 0.3% NaCl), D (30 mg L$^{-1}$ DTA-6 + 0% NaCl), andDS (30 mg L$^{-1}$ DTA-6+ 0.3% NaCl).

tendency to increase and then decrease, as evidenced by a 14.89% and 9.88% increase in AsA in the S treatment compared to the CK treatment 1d and 4d after NaCl stress, and a decrease in AsA in the S treatment compared to the CK treatment 7 d, 10 d, and 13 d after NaCl stress by 15.89%, 13.05%, and 22.83%, respectively. After 1d of NaCl stress, GSH content increased by 6.78% in S treatment compared to CK treatment, and decreased by 43.16%, 33.62%, 40.70%, and 29.96% in S treatment compared to CK treatment AsA after 4 d, 7d, 10 d, and 13 d of NaCl stress, respectively. Under NaCl stress, AS's AsA and GSH contents were higher than those of the S treatment. The AsA and GSH contents of the AS treatment increased by 2.53%-45.65% and 60.17%-126.47%, respectively, compared with the S treatment, while the AsA and GSH contents of DS were significantly higher than those of the S treatment, and the increases of AsA and GSH contents of DS treatment compared with the S treatment ranged from 2.36%–36.61% and 52.90%–99.69%, respectively.

## Effects of PGRs on osmoregulatory substances in rice roots

In the absence of NaCl stress, foliar spraying of both PGRs significantly increased SP and SS contents in rice roots (Fig. 5). The increase in SP and SS content in treatment A over CK ranged from 11.37%–46.96% and 28.76%–114.39%, respectively, while the increase in treatment D over CK ranged from 17.74%–36.13% and 25.33%–125.26%, respectively.

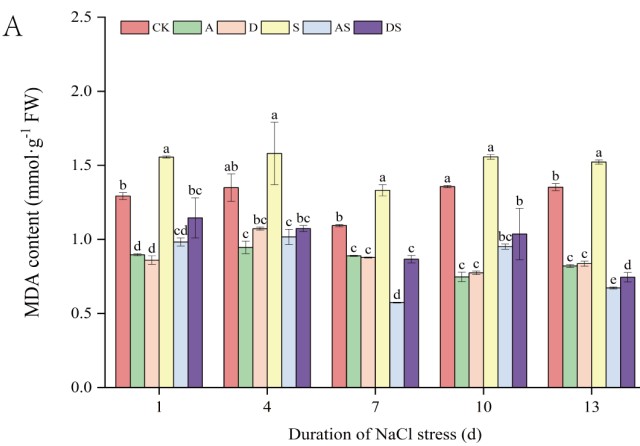

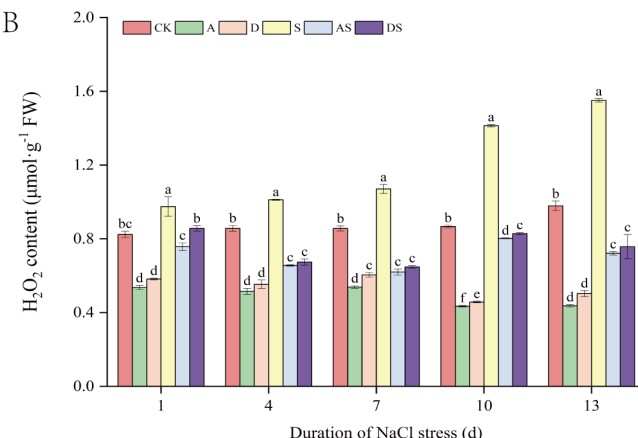

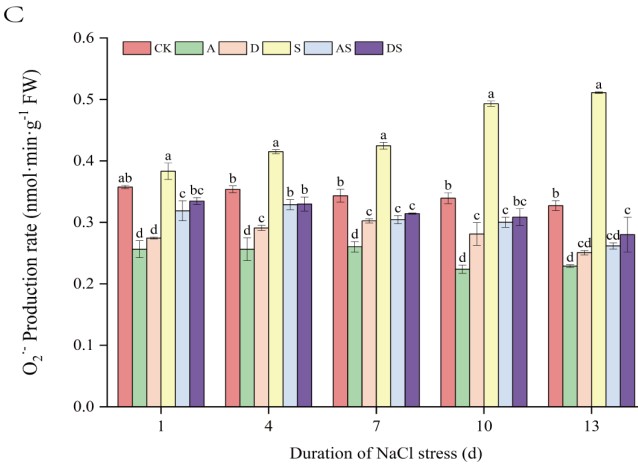

**Figure 3  Effects of PGRs on membrane damage in rice roots.** (A) The content of MDA. Effects of PGRs on the content of MDA 1, 4, 7, 10, and 13 days after NaCl stress. (B) The content of $H_2O_2$. Effects of PGRs on the content of $H_2O_2$ 1, 4, 7, 10, and 13 days after NaCl stress. (c) The content of $O_2 \cdot^-$ Effects of PGRs on the content of $O_2 \cdot^-$ 1, 4, 7, 10, and 13 days after NaCl stress. CK (distilled water + 0% NaCl), S (distilled water + 0.3% NaCl), A (40 mg L$^{-1}$ 5-ALA + 0% NaCl), AS (40 mg L$^{-1}$ 5-ALA + 0.3% NaCl), D (30 mg L$^{-1}$ DTA-6 + 0% NaCl), andDS (30 mg L$^{-1}$ DTA-6 + 0.3% NaCl).

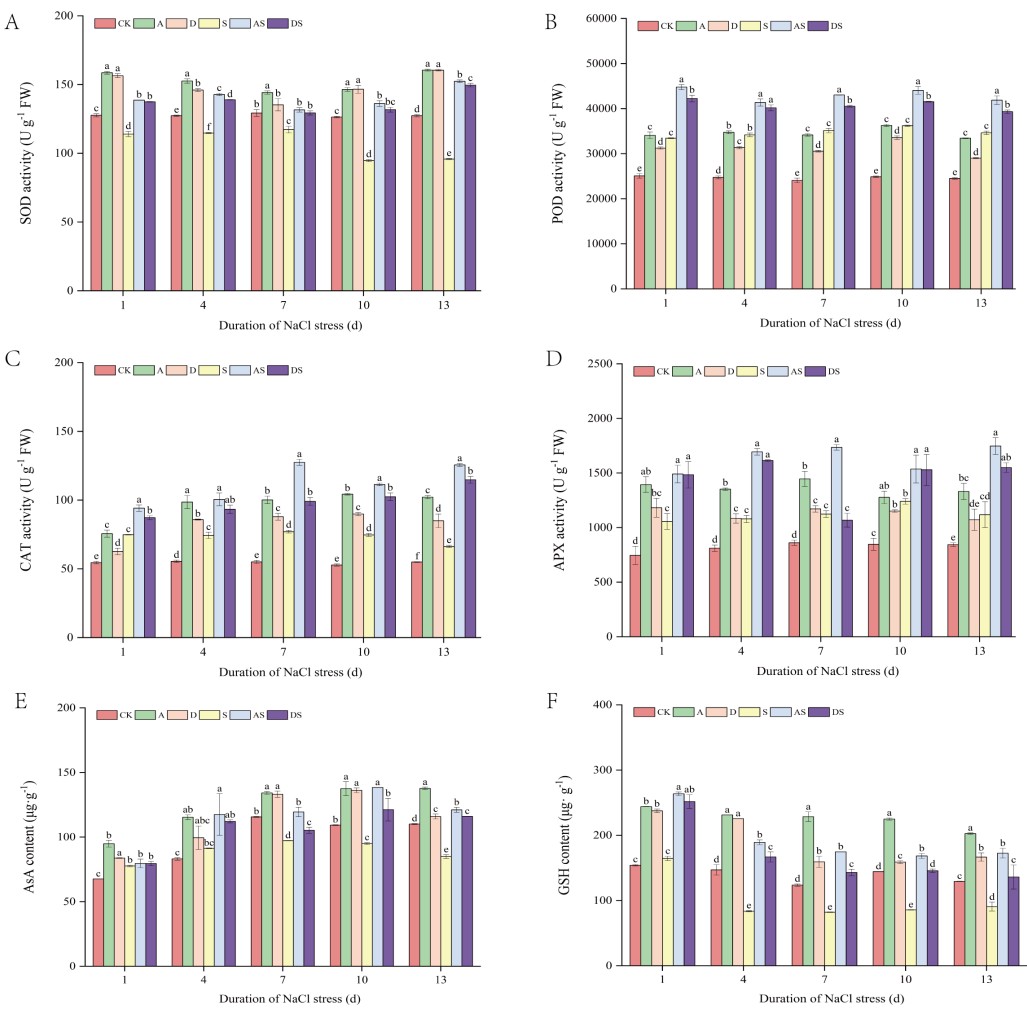

**Figure 4** **Effects of PGRs on antioxidant substances in rice roots.** (A) The activities of SOD. Effects of PGRs on the activities of SOD 1, 4, 7, 10, and 13 days after NaCl stress. (B) The activities of POD. Effects of PGRs on the activities of POD 1, 4, 7, 10, and 13 days after NaCl stress. (C) The activities of CAT Effects of PGRs on the activities of CAT 1, 4, 7, 10, and 13 days after NaCl stress. (D) The activities of APX. Effects of PGRs on the activities of APX 1, 4, 7, 10, and 13 days after NaCl stress. (E) The content of AsA. Effects of PGRs on the content of AsA 1, 4, 7, 10, and 13 days after NaCl stress. (F) The content of GSH. Effects of PGRs on the content of GSH 1, 4, 7, 10, and 13 days after NaCl stress. CK (distilled water + 0% NaCl), S (distilled water + 0.3% NaCl), A (40 mg L$^{-1}$ 5-ALA + 0% NaCl), AS (40 mg L$^{-1}$ 5-ALA + 0.3% NaCl), D (30 mg L$^{-1}$ DTA-6 + 0% NaCl), andDS (30 mg L$^{-1}$ DTA-6+ 0.3% NaCl).

Under NaCl stress, S-treated SP showed an increasing and then decreasing trend compared with CK, while S-treated SS showed an increasing trend compared with CK. The SP content increased by 12.06% and 12.61% at 1 and 4 d of NaCl stress, respectively, and decreased by 15.43%, 18.57%, and 10.21% at 7, 10 and 13 d of NaCl stress, respectively, in the S treatment compared to CK. S treatment increased SS content from 25.79%%–73.52% compared to CK. The SP content of the AS and DS treatments was significantly higher than that of the S treatment from 4 to 13 d after NaCl stress, with increases ranging from 15.02%–48.58% and 7.24%–33.58%, respectively; Under NaCl stress, the SS content of

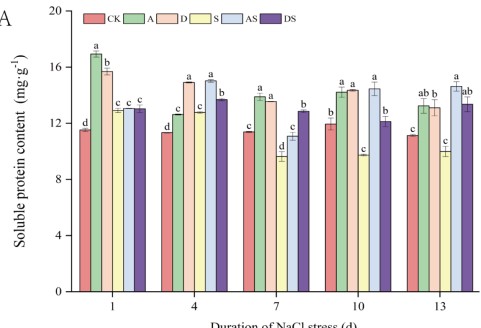
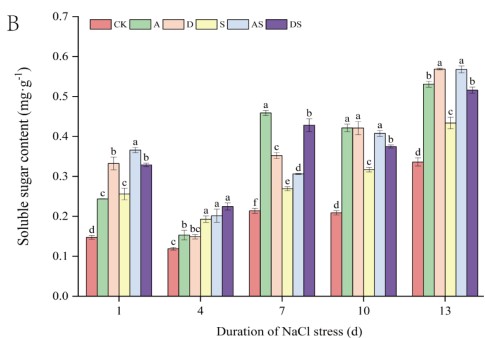

**Figure 5** **Effects of PGRs on soluble protein and soluble sugar content in rice roots.** (A) The content of soluble protein (SP). Effects of PGRs on the content of SP 1, 4, 7, 10, and 13 days after NaCl stress. (B) The content of soluble sugar (SS). Effects of PGRs on the content of SS 1, 4, 7, 10, and 13 days after NaCl stress. CK (distilled water + 0% NaCl), S (distilled water + 0.3% NaCl), A (40 mg L$^{-1}$ 5-ALA + 0% NaCl), AS (40 mg L$^{-1}$ 5-ALA + 0.3% NaCl), D (30 mg L$^{-1}$ DTA-6 + 0% NaCl), and DS (30 mg L$^{-1}$ DTA-6 + 0.3% NaCl).

AS and DS treatments was significantly higher than that of the S treatment, with increases ranging from 4.49%–42.97% and 16.53%–58.99%, respectively. The above results indicated that 5-ALA and DTA-6 could promote the production of osmoregulatory substances in rice roots and maintain cellular homeostasis. 5-ALA was superior to DTA-6 in promoting the production of osmoregulatory substances in roots.

## Effect of PGRs on root hormones in rice

NaCl stress decreased the contents of GA3, JA, IAA, SA, and ZR and increased the contents of ABA in rice roots (Table 2). Both 5-ALA and DTA-6 modulated the effects of NaCl stress on rice root hormones, with 5-ALA being superior to DTA-6 in reducing hormone toxicity under NaCl stress and promoting hormones under non-NaCl stress. Foliar spraying of both PGRs increased the contents of GA3, JA, IAA, SA, and ZR in rice roots under no NaCl stress. The GA3, JA, IAA, SA, and ZR contents were significantly increased by 40.92%, 18.44%, 26.82%, 15.31%, and 24.70% in the A treatment compared to CK treatment; the GA3, JA, IAA, SA, and ZR contents were significantly increased by 21.98%, 8.72%, 15.56%, 13.29%, and 11.82%, respectively; and the A treatment and D treatment reduced ABA content compared with the CK treatment by 12.61% and 6.00%, respectively. Foliar spraying of both PGRs increased GA3, JA, IAA, SA, and ZR content in rice roots under NaCl stress. The AS treatment significantly increased GA3, JA, IAA, SA, and ZR content by 50.83%, 56.12%, 44.18%, 25.50%, and 26.02% compared to the S treatment. The DS treatment significantly increased the GA3, JA, IAA, SA, and ZR content by 24.33%, 26.27%, 20.22%, 16.03%, and 18.15%, respectively, compared to the S treatment; and the contents of ABA in AS and DS treatments were significantly reduced by 26.62% and 18.36%, respectively, compared with the S treatment.

**Table 2  Effects of GPRs on endogenous hormones in rice roots under NaCl stress.**

| Treatment | GA3 (pmol g$^{-1}$) | JA (pmol g$^{-1}$) | IAA (nmol g$^{-1}$) | SA (ng g$^{-1}$) | ZR (ng g$^{-1}$) | ABA (ng g$^{-1}$) |
|---|---|---|---|---|---|---|
| CK | 0.61 ± 0.02d | 11.44 ± 0.13e | 0.25 ± 0.01de | 50.10 ± 0.98c | 45.43 ± 0.98c | 403.38 ± 8.19b |
| A | 0.86 ± 0.00a | 13.55 ± 0.08b | 0.32 ± 0.01ab | 57.77 ± 0.57b | 56.65 ± 0.75a | 352.49 ± 15.95cd |
| D | 0.75 ± 0.01b | 12.44 ± 0.17c | 0.29 ± 0.01bc | 56.76 ± 0.83b | 50.80 ± 1.51b | 379.20 ± 9.43bc |
| S | 0.55 ± 0.01e | 9.47 ± 0.15f | 0.23 ± 0.00e | 49.67 ± 0.63c | 43.29 ± 0.53c | 451.47 ± 6.28a |
| AS | 0.83 ± 0.03a | 14.78 ± 0.15a | 0.32 ± 0.01a | 62.33 ± 0.57a | 54.55 ± 0.17a | 331.28 ± 2.08d |
| DS | 0.69 ± 0.02c | 11.95 ± 0.10d | 0.27 ± 0.00cd | 57.63 ± 0.64b | 51.15 ± 0.23b | 368.59 ± 0.68cd |

**Notes.**

Data represents the means ± standard error. Different lowercase letters in the same column indicate a significant difference in Duncan at P<0.05. CK (distilled water + 0% NaCl), S (distilled water + 0.3% NaCl),A (40 mg L$^{-1}$ 5-ALA + 0% NaCl) , AS (40 mg L$^{-1}$ 5-ALA + 0.3% NaCl), D (30 mg L$^{-1}$ DTA-6 + 0% NaCl) , DS (30 mg L$^{-1}$ DTA-6+ 0.3% NaCl).

## DISCUSSION

### Effects of PGRs on root morphology and endogenous hormones in rice

Roots absorb water and nutrients from the soil and transfer them to the top of the plant, and play an essential role in plant growth and development (*Sainju, Singh & Whitehead, 2005*). Root length, dry matter mass, and root vigor are the most basic indicators of root growth and development in rice (*Rani & Sukumari, 2013*). As the first organ to feel salt stress, the root system produces signaling molecules to adapt to salt stress by regulating the activity of antioxidant enzymes and the expression of related proteins in the cells (*Li et al., 2023*). In this study, we found that root length, root dry matter accumulation, and root vigor of rice showed a decreasing trend with an increase of NaCl stress time, indicating that NaCl stress inhibited the growth of rice (Table 1 and Fig. 2). This may be because NaCl stress increases the concentration of soluble salts in the soil, leading to dehydration of root cells and osmotic stress, which affects the normal metabolism of rice, slows growth, and leads to a reduction in root biomass (*Chen et al., 2016*). 5-ALA and DTA-6 significantly alleviated the inhibition of root growth by NaCl stress and enhanced root-soil interactions and nutrient uptake. This may be due to the fact that 5-ALA and DTA-6 regulate the components of auxin transport under salt stress and promote the growth of rice roots by improving auxin transport. This is similar to previous studies that found that 5-ALA and DTA-6 could promote plant growth under abiotic stresses (*Akram & Ashraf, 2013a*; *Hassan et al., 2022*).

GA3, JA, IAA, and SA are plant hormones that have signaling effects and can improve salt tolerance in rice (*Sakya, Purnomo & Bima, 2022*; *Rady et al., 2021*; *Ahmad et al., 2019*; *Khan et al., 2019*; *Kim, Kim & Shim, 2017*). The phytohormone ABA plays a vital role as a stress hormone in abiotic stress tolerance in plants (*Zhao et al., 2020a*). Phytohormones do not act singly, but instead multiple hormones regulate plant growth and development through synergistic interactions and antagonism. The present study showed that NaCl stress decreased the GA3, JA, IAA, SA, and ZR content and increased the ABA content in rice roots (Table 2). This is similar to a previous study, which found that NaCl stress decreased the GA3 and IAA content and increased the ABA content in maize roots, leading to a decrease in the ratio of type hormones to inhibitory type hormones (*Jin et al., 2023*).

Under salt and non-salt stress, both 5-ALA and DTA-6 increased the GA3, JA, IAA, SA, and ZR content and decreased the ABA content in the root system. These results indicated that both exogenous 5-ALA and DTA-6 could regulate the content of endogenous hormones in rice roots, promote more cell division, clear intracellular ROS, and trigger the defense system, thereby improving the salt tolerance of rice. Interestingly, the regulatory effect of 5-ALA on hormones was better than that of DTA-6, and it was speculated that 5-ALA had a greater impact on the signaling mode, so more effectively regulated the growth and metabolism of rice roots. Exogenous IAA has been reported to promote seed germination and seedling growth by down-regulating the unfavourable endogenous hormone ABA and up-regulating the synthesis of favourable endogenous hormones in cotton (*Zhao et al., 2020c*), which is in agreement with our findings.

## Effects of PGRs on root membrane damage and antioxidant system in rice

NaCl stress disrupts the dynamic balance between ROS production and scavenging in plants. It induces excessive accumulation of ROS in plant cells, leading to impairment of the antioxidant system and exacerbation of membrane lipid peroxidation. These effects arise from their interaction with macromolecules such as lipids, proteins, and nucleic acids (*Alam et al., 2019*). MDA is an important indicator of membrane lipid peroxidation, which reflects the extent of damage to plant membranes under unfavorable environmental conditions (*Nishihara et al., 2003*). $H_2O_2$ is a key signaling molecule in plants that reflects cell senescence and plant resistance responses to environmental stresses (*Huang et al., 2016*). The results of this study showed that NaCl stress increased MDA, $H_2O_2$, and $O_2\cdot^-$ content in rice roots, and exogenous sprays of both 5-ALA and DTA-6 reduced MDA, $H_2O_2$ and $O_2\cdot^-$ content (Fig. 3). The two regulators reduced MDA content by more than 20%, which alleviated salt stress to a certain extent. This is consistent with 5-ALA increasing plant tolerance by enhancing ROS scavenging capacity under abiotic stresses (*Farid et al., 2018*; *Wu et al., 2018*). 5-ALA was more effective than DTA-6 in alleviating root membrane damage. This may be because 5-ALA improves the vigor of rice roots, promotes the entry of water into the cells, and reduces the concentration of sodium ions, thereby reducing the damage to the root cell membrane.

Antioxidant defenses, including enzymatic and non-enzymatic systems, are used to balance the intracellular production of ROS under stress (*Alam et al., 2019*). SOD, POD, APX, and CAT are major enzyme scavengers in plants (*Racchi, 2013*). SOD enzymes are critical enzymes in the antioxidant system of plants, and scavenge superoxide anions, reacting to produce $H_2O_2$ and $O_2$, and CAT and APX catalyze the subsequently produced $H_2O_2$ to produce water and divalent oxygen (*Wang et al., 2019*). POD catalyzes substrate oxidation using $H_2O_2$ as an electron acceptor, and it is considered to be one of the two major indicators of plant stress tolerance along with SOD (*Shu et al., 2017*). In the current study, it was found that SOD activity was reduced by NaCl stress (Fig. 4A), which may be because ROS generated by high NaCl stress could not be scavenged efficiently by antioxidant enzymes, resulting in oxidative damage. POD, APX, and CAT activities were significantly increased under salt stress (Figs. 4B–Figs. 4D), which was consistent with

previous studies that found that sorghum seedlings employ stress responses to avoid large ROS accumulation and increase cell membrane stability (*Yin et al., 2016*). Under NaCl stress, both 5-ALA and DTA-6 enhanced antioxidant enzyme activities and scavenging of reactive oxygen radicals in rice roots, thereby reducing oxidative damage caused by salt stress (Fig. 4). The increase of 5-ALA was higher than that of DTA-6, which may be related to the fact that 5-ALA is the promoter of antioxidant enzyme activity (*Kanto et al., 2015*). It suggests that 5-ALA alleviates salt stress by increasing antioxidant enzyme activities and reducing membrane damage.

AsA, also known as vitamin C, is a small-molecule antioxidant and an electron donor in redox reactions (*Yabuta et al., 2002*). GSH, an essential antioxidant in the AsA-GSH cycle, regulates $H_2O_2$ and influences AsA regeneration. GSH activates the plant defense machinery by participating in redox signaling (*Hojati et al., 2011*), and the redox state of GSH affects crop adaptation to abiotic stresses and largely determines the efficacy of AsA (*Selote & Khanna-Chopra, 2006*). ASA and GSH are two fundamental antioxidants in plant cells that play different and important roles in scavenging free ROS. This may be due to the different intracellular compartments of the two molecules and the different ratios of reduced and oxidized forms under various abiotic stress conditions. In the present study, it was found that NaCl stress increased and then decreased the AsA and GSH contents of rice roots (Figs. 4E and 4F), which may be due to the oxidative stress generated by the plant itself under stress and the weakened synthesis capacity of rice in response to excessive AsA consumption to cope with the damage under high NaCl stress. Under NaCl stress, foliar sprays of both 5-ALA and DTA-6 increased AsA and GSH content, implying that more electron donors are involved in the $H_2O_2$ reduction reaction and promotion of AsA regeneration. This is consistent with the findings that ALA reduces damage caused by zone stress in poplar by increasing AsA and GSH levels (*Castagna et al., 2015*).

Osmotic stress is the main stress in crop damage caused by NaCl stress, which can be alleviated by increasing osmoregulatory substances in the plant to balance vesicular and cytoplasmic water potentials (*Azevedo Neto, Prisco & Gomes-Filho, 2009*). Soluble proteins are important osmoregulatory substances in plants that increase the water retention capacity of cells, thereby protecting biofilms. In the present study, we found that NaCl stress increased and then decreased the soluble protein content (Fig. 5A), which may be due to the protective mechanism in rice leaves in the early stage of NaCl stress that increased the protein content first. The soluble protein contents then decreased in the later stage with the decrease in the water absorption capacity of rice under NaCl stress (*Mahmoud et al., 2019*). Under NaCl stress, foliar sprays of both 5-ALA and DTA-6 increased the soluble protein content in the root system. Increased soluble protein content implies an increase in metabolism-related enzyme proteins, which promotes root growth. Soluble sugars increase the cytoplasmic concentration and provide a carbon source for the plant, participate in the intracellular osmoregulation mechanism, control the water and osmotic potentials, and further enhance the plant's resistance to stress (*Boriboonkaset et al., 2013*). In the present study, we found that NaCl stress increased soluble sugar content, similar to its increase in drought in previous studies (*Diaz-Mendoza et al., 2016*; *Sade et al., 2018*). The results indicated that rice roots could regulate cell osmotic potential by increasing the soluble

protein and soluble sugar contents, improve the ability of cells to absorb and retain water, and protect the vital substances and biofilms of cells by increasing the soluble protein and soluble sugar contents. Under NaCl stress, foliar sprays of both 5-ALA and DTA-6 increased the soluble sugar content in the root system (Fig. 5B), thereby increasing the osmoprotective effect of the rice root system, which ultimately favoured the enhancement of the tolerance of the rice root system to salt stress. These results indicated that 5-ALA and DTA-6 could increase the activities of some enzymes in the salt-resistant sugar and protein biosynthesis pathways, promote the synthesis of sugars and proteins, and thus improve the salt resistance of rice plants.

## CONCLUSION

5-ALA and DTA-6 alleviated salt stress in rice roots in various ways. According to our findings, NaCl stress inhibited rice root growth. Exogenous leaf sprays of 5-ALA and DTA-6 could increase the activity of antioxidant enzymes, increase the content of non-enzymatic antioxidants AsA and GSH in roots, and increase the content of osmotic regulators in roots, thereby reducing the production of MDA, $H_2O_2$, and $O_2^-$, alleviating oxidative damage caused by salt stress, and improving the stability of the root membrane. In addition, the two PGRs increased the content of endogenous hormones GA3, JA, IAA, SA, and ZR and decreased the content of ABA, which enhanced the physiological response and adaptation to NaCl stress and improved the salt tolerance of rice. In conclusion, exogenous 5-ALA and DTA-6 promoted the development of rice roots under salt stress by regulating the antioxidant process and the degree of membrane damage, and exogenous 5-ALA had a better regulatory effect on salt damage in rice roots than exogenous DTA-6. This finding further refines the potential mechanism of PGRs in improving salt tolerance in rice roots. This experiment provides a scientific basis for the optimal regulation of salt resistance in agricultural production from the physiological aspect, but the molecular mechanisms involved in salt resistance optimization of these two regulators needs to be further explored.

## ACKNOWLEDGEMENTS

We are very grateful to all authors for their contributions to this article. We would also like to thank the editor and reviewers for their positive comments.

### Funding

This work was supported by program for the Scientific Research Start-up funds of Guangdong Ocean University (060302052010), the Innovation Team Project of ordinary colleges of the Educational Commission of Guangdong Province (2021KCXTD011), the Research start-up project of Guangdong Ocean University (R20046), and the Research start-up project of Guangdong Ocean University (060302052012). The funders had no role in study design, data collection and analysis, decision to publish, or preparation of the manuscript.

## Grant Disclosures

The following grant information was disclosed by the authors:

Scientific Research Start-up funds of Guangdong Ocean University: 060302052010.

Innovation Team Project of ordinary colleges of the Educational Commission of Guangdong Province: 2021KCXTD011.

Research start-up project of Guangdong Ocean University: R20046.

Research start-up project of Guangdong Ocean University: 060302052012.

## Competing Interests

The authors declare there are no competing interests.

## Author Contributions

- Yaxin Wang conceived and designed the experiments, performed the experiments, analyzed the data, authored or reviewed drafts of the article, and approved the final draft.
- Li-ming Zhao conceived and designed the experiments, prepared figures and/or tables, authored or reviewed drafts of the article, and approved the final draft.
- Naijie Feng conceived and designed the experiments, authored or reviewed drafts of the article, and approved the final draft.
- Dianfeng Zheng performed the experiments, authored or reviewed drafts of the article, and approved the final draft.
- Xue Feng Shen performed the experiments, authored or reviewed drafts of the article, and approved the final draft.
- Hang Zhou performed the experiments, analyzed the data, authored or reviewed drafts of the article, and approved the final draft.
- Wenxin Jiang analyzed the data, prepared figures and/or tables, and approved the final draft.
- Youwei Du analyzed the data, prepared figures and/or tables, authored or reviewed drafts of the article, and approved the final draft.
- Huimin Zhao analyzed the data, prepared figures and/or tables, authored or reviewed drafts of the article, and approved the final draft.
- Xutong Lu analyzed the data, prepared figures and/or tables, authored or reviewed drafts of the article, and approved the final draft.
- Peng Deng analyzed the data, prepared figures and/or tables, authored or reviewed drafts of the article, and approved the final draft.

## Data Availability

The raw measurements are available in the Supplementary Files.

## Supplemental Information

Supplemental information for this article can be found online at http://dx.doi.org/10.7717/peerj.17068#supplemental-information.

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
