# Peer review of "Plant growth regulators mitigate oxidative damage to rice seedling roots by NaCl stress"

_PeerJ, doi:10.7717/peerj.17068_

## Round 0.1 · original submission · Minor Revisions

In addition to reviewers' recommendations;

Lines 104 and 108 may remain the same. I disagree with reviewer 1. But the choice is yours.

Line 114: use the word "enzymatic" instead of "physiological".

You wrote GPRs instead of PGRs in many places, correct them.

Line 223: what was the purpose of correlation analysis? You did not write it in the statistical analysis section of the materials and methods section.

I recommend you rewrite the conclusion section. It is important that you present your recommendations by giving a brief summary of your results.

**Language Note:** The review process has identified that the English language must be improved. PeerJ can provide language editing services - please contact us at copyediting@peerj.com for pricing (be sure to provide your manuscript number and title). Alternatively, you should make your own arrangements to improve the language quality and provide details in your response letter. – PeerJ Staff

Reviewer 1 ·

Basic reporting

1. The sentence construction and English of the manuscript is poor
2. The background information on the study, hypothesis, and research gaps are insufficient
3. Need to rearrange the figures
4. Need to discuss more on the presented results rather than discussing available literature

Experimental design

1. Research findings fall well under the scope of the journal
2. However, need clear presentation of the materials and methods as suggested in the PDF
3. Details on how root sampling was done need to be explained

Validity of the findings

1. Need to highlight, how it benefits growers
2. Where these results could be used
3. Conclusion session needs a thorough revision

Additional comments

Overall, the abstract is good but it could be presented in a more logical structure. It should follow the structure as follows;
1. Highlight the background of the study and research importance
2. Write materials and methods including design of study and location
3. Mention the significant findings, in quantitative terms like % increase or decrease, or values in ()
4. Conclusion and practical utility of the results.

Annotated reviews are not available for download in order to protect the identity of reviewers who chose to remain anonymous.

Reviewer 2 ·

Basic reporting

• The language of the manuscript is easy to comprehend.
• The abbreviation of “Plant Growth Regulators” is written as “PGRs” in the text and “GPRs” in the titles. Please uniform this abbreviation.
• Lines 186 and 188: The letter “d” used as the abbreviation for "day" should be written in lowercase.
• References in the text and the reference list should be handled with greater care. Especially the reference list, should be revised in accordance with the Journal's guidelines.
o Line 47, 107, 115, 119, 120, 121, 122, 123, 124, 125
o Line 381, 391, 393, 405, 418, 421, 428, 436, 446, 452, 455, 461, 466, 472, 474, 478, 488, 504, 506, 512, 515, 518, 521, 528, 534, 540, 542, 545, 548, 551, 554, 557, 560, 566, 569, 581, 583, 586, 589, 592, 606

Experimental design

• Lines 121-122: Was Nakano and Asada (1980) method used to determine the amount of hydrogen peroxide or activity of APX?
• Line 129: What was the wavelength use for SS content determination? 10?
• Line 123-131: Please indicate the standarts used to calculate the SP, SS and GSH content.
• Which method was used to determine the amount of ASA?

Validity of the findings

• Every underlying data set is available. They are controlled, reliable, and sound statistically.
• All results have been adequately discussed.

---

## Round 0.2 · accepted · Accept

Congratulations. After all corrections, your manuscript can be accepted for publication.

Reviewer 1 ·

Basic reporting

The authors have addressed the comments

Experimental design

The authors have mentioned the experimental design

Validity of the findings

-

Additional comments

The authors have addressed the comments, however, there is still a scope for improvement of the article structure.
Revision is ok, however, there is still a scope to concise the content, word count is 368, but it could be reduced to 300.

Annotated reviews are not available for download in order to protect the identity of reviewers who chose to remain anonymous.

Reviewer 2 ·

Basic reporting

*The references in the text and the reference list have been updated, and
* All of my spelling correction recommendations have been implemented.
* Figures are rearranged

Experimental design

*Confusion regarding methodology has been resolved.

*How the analyzes are made is expressed more clearly and understandably.

Validity of the findings

Every underlying data set is available. They are controlled, reliable, and sound statistically.